## Overview Review

firearm suicide; socio-ecological model; Black and Latino; adolescents; United States

**Corresponding author:**
Dasha J. Rhodes;
Email: dasharhodes@usf.edu

# Intersecting harms: A structural perspective on firearm suicide among black and Latino youth in the United States

Dasha J. Rhodes[1] , Sharon Gandarilla-Javier[2], Kevin Antwi[3] and Nuavia Stewart[1]

[1]School of Social Work, University of South Florida College of Behavioral and Community Sciences, Sarasota, FL, USA; [2]Department of Law, Police Science, and CJA, John Jay College of Criminal Justice, New York, NY, USA and [3]School of Social Work, University of South Florida College of Behavioral and Community Sciences, Tampa, FL, USA

## Abstract

Firearm suicide among adolescents in the United States has increased in recent years, with Black and Latino youth experiencing disproportionately rising rates. Although firearm violence and mental health disparities have received growing attention, the structural conditions that shape racial inequities in firearm suicide risk remain insufficiently examined. This overview applies an intersectional and structural lens to analyze how systemic inequities including residential segregation, concentrated disadvantage, punitive school discipline practices, underinvestment in mental health infrastructure and commercial determinants of firearm availability contribute to differential suicide risk. Drawing on interdisciplinary literature and recent epidemiologic data, the manuscript maps the causal pathways through which structural racism and institutional inequities shape exposure to community violence, access to culturally responsive care, crisis response practices and household firearm environments. It further examines how these mechanisms interact with gender and lethal means availability to amplify disparities in suicide mortality. The analysis underscores the importance of multi-level, upstream interventions that address structural inequities, strengthen community-based supports and reduce access to lethal means. By reframing firearm suicide as a structurally patterned outcome rather than an individual-level phenomenon, this work advances a socio-ecological understanding of adolescent suicide prevention with implications for structural reform and suicide prevention policy.

## Impact statement

Firearm suicide is an escalating public health concern in the United States, with recent increases disproportionately affecting Black and Latino adolescents. This manuscript reframes firearm suicide beyond individual behavior by situating it within intersecting structural determinants, including racialized segregation, concentrated disadvantage, inequitable mental health access, commercial influences on firearm availability and punitive institutional responses to youth distress. By mapping how these systemic forces converge to shape exposure to violence, crisis response and lethal means access, this work highlights why suicide disparities persist across racial and ethnic groups. The manuscript contributes to global mental health scholarship by demonstrating how structural and commercial determinants interact to influence suicide risk, offering a framework applicable to diverse sociopolitical contexts even where dominant means differ. Rather than centering race or socioeconomic status as isolated risk factors, the analysis emphasizes the systems that produce vulnerability and calls for prevention strategies grounded in structural reform, equity and culturally responsive care.

## Introduction

Gun-related suicides among adolescents in the United States represent an escalating public health crisis that disproportionately affects Black and Latino(a) youth (Robinson et al., 2022; Talley et al., 2022). Despite growing national discourse on gun control, mental health and community violence, the intersection of structural racism, firearm access and suicide remains underexplored. Black and Latino adolescents, particularly those in economically marginalized communities, face heightened vulnerability due to systemic inequities, racialized violence and limited access to culturally responsive mental health care (Formica, 2021; Alvarez et al., 2022).

Emerging epidemiologic trends indicate a racial shift in suicide burden, with firearm suicide rates rising sharply among Black and Latino adolescents in recent years. These patterns challenge historically dominant narratives that framed suicide as disproportionately affecting White youth and underscore the need to examine the structural and institutional conditions through which

risk is produced and sustained. Rather than reflecting isolated individual pathology, these shifts suggest that broader social, economic and policy environments are shaping differential exposure to violence, access to care and access to lethal means.

While suicidal ideation occurs across racial and ethnic groups, access to firearms significantly increases the likelihood that impulsive behaviors result in fatal outcomes. Many Black youth who die by firearm suicide have no prior documented mental health diagnosis but live in environments characterized by violence, grief and systemic neglect, suggesting that risk is often shaped less by individual pathology than by structural conditions (Talley et al., 2022). Although this manuscript focuses on Black and Latino adolescents in the United States, the broader structural mechanisms described are not confined to one national context. Across global settings, marginalized youth experience elevated suicide risk in environments shaped by discrimination, violence exposure and inequitable access to care. What distinguishes the U.S. context is the widespread availability of firearms as a lethal means. This review applies a socio-ecological and structural lens to examine how systemic inequities converge to shape firearm suicide risk among Black and Latino youth and advances a global mental health perspective that centers structural reform and equity-focused prevention.

## Background

Firearm suicide constitutes a distinct and escalating public health crisis within the broader landscape of firearm-related mortality in the United States. Since the mid-1990s, suicide by firearm has accounted for the majority of firearm deaths and continues to drive national mortality trends (Rosenberg, 2021; Johns Hopkins Center for Gun Violence Solutions, 2025). In 2024, the U.S. Surgeon General declared firearm violence a public health emergency, reinforcing longstanding calls from major medical and public health organizations to address firearm harm through coordinated prevention strategies (Office of the Surgeon General & National Action Alliance for Suicide Prevention, 2021; Office of the Surgeon General, 2024).

Among adolescents, firearm suicide remains the most lethal method of self-harm, shaped by firearm accessibility, storage practices and timely access to mental health intervention (Weigend-Vargas et al., 2024). Emerging data demonstrate a racial shift in suicide burden. When examining suicide trends by race and ethnicity rather than age alone, recent reports indicate that suicide rates among non-Hispanic Black individuals have increased significantly in recent years, with Hispanic/Latino individuals experiencing the second-highest increase (CDC, 2025). Between 2014 and 2023, firearm suicide rates rose sharply among Black adolescents and increased substantially among Latino adolescents. Most notably, Black adolescents exceeded their White peers in firearm suicide rates for two consecutive years (2022 and 2023), marking a departure from historical patterns in which White youth had higher prevalence (Kim et al., 2025; Price and Khubchandani, 2019).

The Centers for Disease Control and Prevention emphasize that there is no single cause of suicide; rather, shifts in suicide rates reflect the combined influence of individual, relationship, community and societal factors that affect racial and ethnic groups differently (CDC, 2025). Structural racism that operates through historically entrenched policies and institutional practices shapes the distribution of risk and protection across communities (Bailey et al., 2017, 2021). Through residential segregation, differential

investment in public institutions and inequities in healthcare access, structural racism influences neighborhood conditions, exposure to violence and the availability of culturally responsive mental health services.

Black and Latino adolescents disproportionately reside in communities shaped by concentrated disadvantage, environmental stressors and fragmented service infrastructures. These contexts influence exposure to firearm violence, institutional responses to youth distress and access to preventive care. While community violence is not framed as a direct cause of firearm suicide, it operates as a contextual stressor that interacts with limited access to culturally responsive care and permissive firearm environments to heighten vulnerability.

Although firearms are the dominant means of youth suicide in the United States, the structural mechanisms shaping suicide disparities are not unique to this country. Across global settings, marginalized youth experience elevated suicide risk in environments marked by discrimination, socioeconomic exclusion and inequitable access to care. The following sections apply a socio-ecological framework to clarify how these multilevel structural conditions converge to produce differential risk among Black and Latino adolescents.

## Approach and limitations

To provide context for this phenomenon, we conducted a targeted narrative review of peer-reviewed literature, governmental reports and public health data published between 2000 and 2025. Searches were conducted using databases including PubMed, PsycINFO and Google Scholar, supplemented by manual searches of key public health and policy sources. Studies were included if they addressed adolescent firearm access, storage practices, suicidal behavior, violence exposure or structural determinants of firearm injury among racially and ethnically marginalized youth.

This manuscript is a selective narrative synthesis rather than a systematic review. The search strategy prioritized U. S.-based literature and policy documents to support the manuscript's conceptual focus on firearm suicide in Black and Latino adolescents within the U.S. context. As such, it does not represent an exhaustive review of international suicide research.

Several limitations in the existing evidence base warrant consideration. Suicide mortality data are affected by variability in reporting practices, potential misclassification of suicide versus homicide or unintentional injury and limited granularity in race and ethnicity categories. Improvements in surveillance systems over time may partially account for large percentage increases reported across certain periods. Additionally, reliance on secondary data sources and heterogeneous study designs limits causal inference.

Despite these constraints, the narrative approach allows for integration of interdisciplinary scholarship to advance a structural and socio-ecological framework. By using the U.S. context as an illustrative case since firearms represent the dominant means of youth suicide, this review offers conceptual insights applicable to other high-inequality settings globally, even where the dominant means of suicide differ.

## Conceptual framework

To organize the multilevel drivers of firearm suicide among Black and Latino youth, a socio-ecological model (SEM) was applied.

The SEM moves beyond individual-level pathology and situates suicide risk within nested structural, institutional, community and individual contexts. Within this framework, structural racism functions as a fundamental cause that shapes the distribution of risk and protective factors across levels (Alvarez et al., 2022; Gillum et al., 2025).

Rather than treating determinants as isolated factors, the SEM clarifies how structural conditions cascade downward to influence institutional practices, neighborhood environments and individual vulnerability. Structural determinants such as residential segregation, economic disinvestment and firearm policy environments shape institutional responses including school discipline practices, crisis intervention models and healthcare access. These institutional and policy arrangements influence community-level exposure to firearm violence, social norms surrounding firearm presence and availability of culturally responsive mental health services. In turn, these layered conditions shape individual experiences of trauma, help-seeking behavior and access to lethal means.

Table 1 categorizes these determinants across SEM levels, distinguishing structural, institutional, community and individual contributors to firearm suicide risk. The table emphasizes that while determinants operate at different levels, they are interconnected rather than independent.

Figure 1 illustrates the directional relationships among these levels, highlighting the causal pathway from structural inequities to differential exposure to violence, cumulative psychological burden, unequal access to care and increased lethality when firearms are accessible. This visualization underscores that firearm suicide among Black and Latino adolescents is not the product of isolated behavior, but the predictable outcome of interacting structural and institutional forces. Moreover, organizing the evidence within this framework clarifies how multilevel inequities converge to produce disparities in firearm suicide and provides a foundation for targeted, equity-focused intervention strategies discussed in subsequent sections.

## Linking structural racism and inequities to suicide risk

Building on the socio-ecological framework outlined above, structural inequities operate through interrelated pathways that shape firearm suicide risk among Black and Latino adolescents. Structural racism influences the distribution of neighborhood resources, institutional responses and policy environments, shaping exposure to violence, access to care and access to lethal means (Bailey et al., 2017, 2021).

First, residential segregation and sustained economic disinvestment concentrate poverty and limit institutional resources in historically marginalized communities. These structural conditions increase exposure to community violence and chronic stress while reducing access to stabilizing supports. Second, institutional practices that include punitive school discipline policies and law enforcement–led crisis responses may criminalize expressions of distress among Black and Latino youth rather than facilitating timely mental health intervention. These practices increase justice system contact and undermine trust in help-seeking systems, delaying or preventing access to care (Alvarez et al., 2022). Third, structural underinvestment in culturally responsive mental health services constrains early identification and sustained treatment. Disparities in insurance coverage, workforce representation and geographic availability reduce the likelihood that Black and Latino adolescents receive timely intervention, particularly during acute crises. And, finally, firearm policy environments and household firearm availability shape the lethality of suicidal behavior. While suicidal ideation occurs across racial and ethnic groups, access to highly lethal means significantly increases the probability that attempts result in death. In this way, structural inequities interact with firearm access to amplify fatality risk rather than uniformly increasing suicidal ideation.

Furthermore, it is important to distinguish firearm suicide from firearm homicide trends. Although both outcomes are shaped by shared structural determinants, firearm homicides are disproportionately concentrated in urban neighborhoods experiencing chronic disinvestment, whereas firearm suicides are often more prevalent in rural and suburban contexts characterized by higher household firearm ownership and geographic isolation (Fontanella et al., 2015; Hemenway, 2017). National data consistently show that youth firearm suicide rates are substantially higher in rural areas than in urban areas (Fontanella et al., 2015). Emerging evidence suggests that these rural–urban patterns intersect with race and ethnicity in complex ways, with Black and Latino youth experiencing vulnerability in both contexts due to distinct but overlapping structural mechanisms. Together, these pathways illustrate how structural racism functions as an organizing force that shapes institutional practice, community exposure and access to lethal means, converting layered inequities into differential suicide risk.

## Challenges in suicide data reporting and classification

Interpretation of suicide trends must account for structural and methodological limitations in injury mortality surveillance systems. Research shows that suicide-related datasets are affected by variability in reporting practices, misclassification of intent and limited

**Table 1.** Application of the socio-ecological model to categorize structural, institutional and individual drivers of firearm suicide

| Level of influence | Specific determinants | Impact on firearm-related suicide risk |
|---|---|---|
| Structural | • Residential segregation, systemic disinvestment and federal/state firearm legislation. | • Structural racism and residential segregation. <br> • Commercial determinants (firearm industry marketing/availability). <br> • Permissive firearm legislation. |
| Institutional | • Punitive school discipline (suspensions, law enforcement referrals). <br> • Police-led crisis response models. <br> • Chronic underinvestment in local health infrastructure. | • Replaces mental health support with criminalization, increasing trauma and undermining trust in help-seeking systems. |
| Community | • Chronic exposure to community firearm violence. <br> • Social learning and contagion effects. <br> • Rural–urban resource disparities. | • Normalizes firearm presence and generates cumulative community trauma, heightening psychological distress. |
| Individual | • Household firearm access and storage practices. <br> • Cultural stigma and historical mistrust of medical systems. <br> • Gender norms and masculinity. | • Reduces the likelihood of seeking care while increasing the probability that an impulsive crisis results in a fatal outcome. |

**Structural**
- Structural racism
- Historical and ongoing segregation
- Permissive firearm policies
- Underinvestment of mental health services
- Societal stigma and mistrust

**Institutional**
- School discipline and policing
- Justice system contact
- Culturally unresponsive mental health services
- System-level barriers to care

**Community**
- Segregation and neighborhood poverty
- Disinvestment in local resources
- Community violence
- Firearm availability

**Individual**
- Black and Latino Youth
- Trauma exposure
- Criminalization of distress
- Mental health burden and constrained help-seeking
- Greater risk of firearm suicide

**Figure 1.** Conceptual Model of the Socio-Ecological Model and Firearm Suicide Among Black and Latino Adolescents.

granularity in race and ethnicity categories (Rockett et al., 2010, 2021). Determinations of suicide often rely on medicolegal judgment, which may be constrained by incomplete contextual information, especially in cases involving firearms, overdose or ambiguous circumstances.

Rockett et al. (2021) found that misclassification of suicide as unintentional injury or homicide occurs disproportionately among racial and ethnic minority populations, with non-Hispanic Black decedents having significantly higher odds of misclassification compared to non-Hispanic White individuals. Factors contributing to this pattern include stigma surrounding suicide, family reluctance to report self-inflicted harm, institutional bias and differential investigative practices. As a result, national reporting systems such as the CDC web-based inquiry statistics query and reporting system may underestimate the true prevalence of suicide among historically marginalized groups.

Similar concerns emerge in research involving detained and system-involved youth. Abram et al. (2014) documented substantially higher rates of suicidal ideation and behavior among detained adolescents than are reflected in school-based or community surveys. Underreporting in these contexts is shaped by fear of legal consequences, stigma, distrust of authorities and cultural norms discouraging disclosure of emotional distress. Administrative data systems may further obscure subgroup variation by aggregating race and ethnicity categories, limiting the precision of disparity estimates (Rockett et al., 2010; Abram et al., 2014).

These reporting limitations help contextualize large percentage increases in suicide rates observed over time. For example, trends indicating nearly 100% increases across certain periods must be interpreted in light of improvements in surveillance systems, evolving classification standards and reductions in misclassification

(Rockett et al., 2021). Such changes may reflect both true increases in risk and enhanced detection of previously undercounted deaths. However, by acknowledging these constraints does not negate the documented rise in firearm suicide among Black and Latino youth; rather, it underscores the importance of cautious interpretation and continued investment in equitable, accurate mortality surveillance. Strengthening data systems is essential to understanding and addressing structural disparities in suicide risk.

### Community exposure among Latino and black youths

The rising rates of firearm suicide among Black and Latino youth must be understood in relation to disproportionate exposure to community firearm violence and heightened environmental risk for firearm access in segregated and economically disadvantaged communities. Chronic exposure to firearm violence is associated with long-term behavioral health consequences, including post-traumatic stress, depression, anxiety, substance use disorders and elevated suicide risk (Ehrlich et al., 2022; Lennon et al., 2024; Rosenthal et al., 2025; Ranney et al., 2019). These harms are concentrated in neighborhoods shaped by poverty and sustained disinvestment (Krivo et al., 2009; Knopov et al., 2019).

National surveillance data indicate that witnessing community violence is strongly associated with gun carrying, substance use and suicidal behaviors among adolescents, particularly Black and Latino youth (Harper et al., 2023). Past-year exposure to firearm violence can affect up to 80% of adolescents in high-poverty communities, compared with approximately 22% in low-poverty, predominantly White areas (Kravitz-Wirtz et al., 2022). Such disparities reflect uneven exposure to trauma and firearm presence

across neighborhoods, reinforcing how structural conditions shape risk environments.

Within the socio-ecological framework, community exposure functions as an intermediate mechanism linking structural inequities to individual-level vulnerability. Exposure to violence does not directly cause suicide; rather, repeated trauma, grief and normalization of firearm presence can heighten psychological distress while increasing familiarity with and access to lethal means. In settings where culturally responsive mental health services are limited, these cumulative stressors may go unaddressed, increasing the likelihood that acute crises escalate.

Exposure to firearm suicide itself may further intensify risk through social learning and contagion effects. Learning about or witnessing firearm suicide has been associated with increased suicidal ideation and attempts among peers (Hill et al., 2020; Martinez et al., 2023; Swanson and Colman, 2013). Policymakers, including the Congressional Black Caucus Emergency Taskforce on Black Youth Suicide and Mental Health (CBCET, 2019), have recognized the role of community-level exposure and contagion in shaping racialized disparities in youth suicide.

Comparable patterns have been documented outside the United States. In Canada, Indigenous and racialized immigrant youth experience elevated suicide risk associated with community violence, historical trauma and inequitable access to mental health services (Public Health Agency of Canada, 2022). Although firearms are not the dominant means of suicide in many countries, these parallels indicate that disparities in youth suicide are shaped by structural inequities and exposure to violence across contexts. The U.S. case is distinguished by the widespread availability of firearms, which increases the lethality of crises occurring within structurally marginalized environments.

### Barriers to mental health care

As outlined in the socio-ecological framework, institutional access to mental health services represents a critical pathway through which structural inequities translate into differential suicide risk. Access to care remains a significant challenge for Black and Latino adolescents segregated and economically marginalized communities (Williams et al., 2019). Underinvestment in local health systems, workforce shortages and fragmented service delivery reduce the availability of preventive and sustained mental health support (Bailey et al., 2017).

Barriers to care operate across structural, logistical and cultural dimensions. Adolescents from low-income households often encounter transportation limitations (Syed et al., 2013), financial constraints that hinder adherence to treatment plans (Lightfoot et al., 2017) and limited geographic proximity to providers (Gaskin et al., 2011; Vander Wielen et al., 2015). Even when services are technically available, disparities in insurance coverage and provider distribution restrict timely intervention.

Cultural stigma and historical mistrust of medical institutions further shape help-seeking behaviors. In many Black and Latino communities, mental health concerns may be perceived as personal weakness or minimized altogether (Forcén et al., 2023; White, 2019). Experiences of discrimination, neglect and racism within healthcare systems reinforce distrust and discourage disclosure of suicidal thoughts (Kosyluk et al., 2022). Compared to White youth, Black and Latino adolescents are significantly less likely to receive mental health services following suicidal ideation or attempts (Freedenthal, 2007; Malhotra et al., 2015; Bommersbach et al., 2022). Notably, disparities persist even after controlling for socioeconomic status and insurance coverage, underscoring the role of structural racism within

healthcare systems (Bommersbach et al., 2022). Limited racial and ethnic diversity among mental health providers further constrains culturally grounded engagement and trust (Lin et al., 2018).

### School-based institutional responses

Schools often serve as primary points of contact for adolescent mental health services. However, schools serving predominantly Black and Latino students are frequently underfunded and less likely to have adequate staffing of counselors or social workers (Bowen et al., 2025). In these settings, institutional responses to distress may operate through disciplinary systems rather than mental health referral pathways.

Students of color are disproportionately subjected to suspensions, expulsions and law enforcement referrals, while their White peers are more often directed to supportive services for similar behaviors (U.S. Government Accountability Office, 2018). These patterns reflect discriminatory disciplinary practices that criminalize behavioral expressions of distress rather than offering intervention and care. Research indicates that perceived discrimination within school settings contributes to peer victimization, emotional distress and diminished sense of belonging (Seaton et al., 2013). Similarly, perceptions of procedural injustice undermine institutional trust and educational outcomes among Latino youth (Dunning-Lozano et al., 2020). Exclusionary discipline is also associated with depression, hopelessness and elevated risk of suicidal ideation and attempts (McKinnon et al., 2024).

When school-based systems lack sufficient mental health resources, crises may escalate beyond educational institutions and into law enforcement responses. In this way, school discipline policies function as an institutional bridge between unmet mental health needs and justice system involvement, reinforcing rather than mitigating vulnerability.

### Law enforcement in crisis response

Law enforcement frequently becomes the default responder to mental health crises, including those involving suicidal ideation. The reliance on police rather than mental health professionals has been associated with disproportionate harm in communities of color (Anene, 2022). Black and Latino individuals face increased risk of trauma, criminalization and fatal encounters during police interactions.

Individuals with serious mental illness are significantly more likely to experience police involvement (Laniyonu and Goff, 2021), and those with untreated mental illness face substantially higher risk of fatal police encounters, with disproportionate impact among Black Americans (Fuller et al., 2015). Exposure to police violence and high-profile incidents has also been associated with increased suicide risk among Black youth (Carney-Knisely et al., 2024).

Exposure to severe violence, adversity and victimization including police-related encounters has been linked to depression, anger and behavioral health challenges (Turner et al., 2006). DeVylder et al. (2017) found associations between police victimization and increased suicide attempts, underscoring the potential psychological consequences of law enforcement contact. Although much of the literature focuses on adults, these patterns highlight broader institutional dynamics that shape youth vulnerability.

Additionally, several crisis response models have been developed to reduce reliance on police during mental health emergencies. Crisis Intervention Teams provide specialized training to officers in collaboration with mental health professionals (Compton et al., 2014). Co-Responder Teams pair officers with clinicians to facilitate immediate care access, while Mobile Crisis Teams deploy community-based

mental health professionals independently of law enforcement involvement (Compton et al., 2014). These models have demonstrated improved outcomes and reduced harm, particularly in communities of color. However, youth-specific concerns remain. Black and Latino adolescents are more susceptible to negative experiences during police encounters, particularly when compounded by prior exposure to systemic racism (Turney et al., 2022). In school settings, law enforcement involvement in behavioral health crises can further entrench the school-to-prison pipeline and erode institutional trust.

Within the broader structural cascade, reliance on law enforcement as a crisis response mechanism represents a downstream institutional consequence of under-resourced mental health systems. When preventive services are inaccessible and school-based supports are limited, crises escalate to punitive systems rather than therapeutic intervention. These institutional dynamics convert unmet mental health needs into criminal justice involvement, reinforcing trauma and increasing vulnerability among already marginalized youth.

## Determinants of firearm access

Firearm access represents a critical component of the structural cascade linking inequity to suicide lethality. While suicidal ideation occurs across racial and ethnic groups, the availability of highly lethal means significantly increases the probability that attempts result in death (Siegel et al., 2016; Swanson et al., 2021). Evidence consistently demonstrates that most firearms used in youth suicides originate from the adolescent's own home or the home of a family member (Johnson et al., 2010; Cunningham et al., 2018), underscoring the importance of household firearm environments.

National survey data indicate that approximately one-third of U.S. household's own firearms, and many children live in homes where guns are stored loaded and unlocked, increasing the likelihood of youth access (Azrael et al., 2018; Miller and Azrael, 2022). These household patterns are not randomly distributed but are shaped by broader sociopolitical and commercial forces. Research suggests that widespread civilian gun ownership in the United States is reinforced by political and legal structures and influenced by industry practices, positioning firearms as a commercial determinant of health (Maani et al., 2020; Hyder et al., 2021).

Population-based findings from the Youth Risk Behavior Survey further demonstrate that firearm carrying among high school students is significantly associated with exposure to community violence, substance use and suicidal ideation (Simon et al., 2022; Harper et al., 2023). Such findings suggest that firearm access operates within broader environments marked by trauma exposure and behavioral health risk rather than as an isolated household phenomenon.

Structural inequities such as residential segregation, concentrated disadvantage and systemic barriers to safety and services elevate both firearm exposure and firearm-related harm among Black and Latino youth (Bottiani et al., 2021). Such considerations indicate that firearm access among U.S. adolescents is shaped not only by household storage practices and individual attitudes, but also by broader political, commercial and structural conditions that normalize firearm availability and unevenly distribute exposure to lethal means.

## Gender differences and intersectionality

Gender differences in firearm suicide are pronounced during adolescence. Across racial and ethnic groups, boys and young men experience substantially higher rates of firearm suicide than girls and young women, reflecting differences in method selection and exposure to highly lethal means. Research indicates that males are more likely than females to use firearms in suicide attempts (Vander Stoep et al., 2011), and access to firearms in the home further increases fatality risk when guns are stored loaded or unlocked (Swanson et al., 2021). At the population level, states with higher rates of firearm ownership exhibit higher firearm suicide rates among males (Siegel et al., 2016), illustrating how firearm availability intersects with gendered patterns of behavior.

Gender differences in firearm suicide must further be examined in relation to race and structural inequities. Intersectional research demonstrates that suicide risk varies across gendered and racialized social positions shaped by structural discrimination rather than individual characteristics alone (Opara et al., 2022; Forrest et al., 2023). Black and Latino male adolescents may face compounded vulnerability due to racialized surveillance, stigma surrounding mental health disclosure and reduced access to culturally responsive services (DuPont-Reyes et al., 2019; Williams-Butler et al., 2023; Akkas and Corr, 2024). At the same time, Latina adolescents have reported elevated rates of suicidal ideation and attempts, highlighting that suicide risk does not operate uniformly across gender or racial groups (Price and Khubchandani, 2017; Hausmann-Stabile and Gulbas, 2022).

## Implications for structural reform and suicide prevention

Promoting the mental health and well-being of Black and Latino adolescents requires structural reforms that extend beyond individual-level intervention and engage the institutional and policy environments that shape risk. Because firearm suicide emerges from interacting systems of structural inequity, community exposure, institutional response and lethal means availability, prevention strategies must operate across multiple levels simultaneously.

At the structural and policy level, reforms should prioritize equitable investment in mental health infrastructure, safe firearm storage and regulatory policies and the reduction of commercial practices that normalize widespread firearm availability. Strengthening state-level safe storage requirements and supporting public health-oriented firearm policies are essential components of lethal means reduction. These efforts must be implemented alongside policies that address concentrated disadvantage, housing segregation and resource inequities that contribute to cumulative trauma exposure.

At the institutional level, schools, healthcare systems and crisis response agencies should shift from punitive and law enforcement driven models toward restorative, trauma-informed and culturally grounded approaches. Expanding school-based health centers remains important (Bowen et al., 2025), but structural reform must also include improving access to outpatient and community-based mental health services, strengthening workforce diversity (Sadusky et al., 2024) and reducing systemic barriers to care such as insurance gaps and geographic maldistribution of providers. Disciplinary policies that criminalize behavioral distress should transition toward restorative practices that reinforce belonging and accountability without escalating youth into justice systems.

Community-level strategies should invest in trusted local organizations, youth-centered programming and culturally embedded support networks. Partnerships with community leaders and faith-based institutions can enhance credibility and engagement in historically marginalized neighborhoods. In resource limited settings,

task-sharing models and community health worker programs offer scalable approaches to expanding culturally relevant mental health support.

These recommendations align with the World Health Organization's LIVE LIFE framework, a global suicide prevention strategy. The framework emphasizes means restriction, responsible media communication, socioemotional skill-building and early identification and support, offering a coherent set of evidence-based pillars for multilevel suicide prevention across diverse contexts (World Health Organization, 2020).

Within structurally marginalized communities, operationalizing these pillars requires adaptation to local social and political contexts, including addressing workforce shortages, institutional mistrust and financial constraints. Although, this analysis centers on the United States, the structural mechanisms identified here are not unique to one national context. While the dominant means of suicide vary globally, strategies that reduce access to lethal means, expand culturally grounded supports and confront structural inequities remain broadly applicable. Community-based mental health initiatives in low- and middle-income countries and culturally adapted interventions within Indigenous communities illustrate how prevention efforts can be locally tailored while grounded in shared structural principles.

By implementing upstream structural reforms, requires acknowledging persistent barriers, including political resistance, resource limitations and social instability. Without sustained investment and cross-sector collaboration, structural interventions risk remaining aspirational rather than actionable. Embedding suicide prevention within broader efforts to advance health equity and institutional accountability is therefore critical to long-term sustainability.

## Conclusion

The observed decrease in firearm suicide among White youth may reflect comparatively greater structural protections, including higher rates of private insurance coverage, greater access to specialty mental health services, treatment of suicidality as a clinical rather than criminal concern and household-level differences in firearm storage practices and service navigation. These explanations warrant cautious interpretation and further empirical examination. In contrast, Black and Latino adolescents often experience cumulative exposure to community violence, under-resourced educational systems, institutional mistrust and punitive responses to behavioral distress. When firearms are accessible within these environments, the lethality of crisis is amplified. Examining these structural contrasts clarifies why trends in firearm suicide diverge across racial and ethnic groups.

A socio-ecological perspective underscores that adolescent mental health cannot be disentangled from the broader structural conditions in which youth are embedded. Structural racism, commercial determinants of firearm availability, institutional underinvestment and fragmented systems of care interact to shape differential exposure to both distress and lethal means. Addressing firearm suicide among Black and Latino adolescents therefore requires interventions that extend beyond individual-level treatment and confront the systemic drivers of inequity.

Although this analysis centers on the United States, the underlying mechanisms of commercial influence, concentrated disadvantage, institutional mistrust and unequal access to care extend beyond national boundaries. While the dominant means of suicide differ globally, the interaction between structural inequities and lethal means availability remains a central consideration for prevention efforts. Integrating structural and commercial determinants into suicide prevention research and policy can inform strategies that are culturally grounded, context-specific and adaptable across diverse settings.

Reducing disparities in firearm suicide among Black and Latino youth ultimately depends on sustained structural reform, cross-sector collaboration and accountability within education, healthcare, public health and policy systems. Reframing firearm suicide as a structural phenomenon rather than an individual failing shifts the focus toward systemic solutions capable of advancing equity and strengthening adolescent mental health both within and beyond the United States

**Open peer review.** To view the open peer review materials for this article, please visit http://doi.org/10.1017/gmh.2026.10208.

**Data availability statement.** No datasets were generated or analyzed during the current.

**Acknowledgments.** A generative artificial intelligence tool (ChatGPT, OpenAI, accessed January–February 2026) was used to assist with language refinement and structural organization during manuscript revision. The tool was not used for data analysis or to generate original research ideas. All substantive content was originally developed, reviewed and approved by the authors.

**Author contribution.** Dasha Rhodes led the conceptualization, framing and writing of the original draft. Sharon Gandarilla-Javier contributed to the policy and education sections and participated in manuscript editing. Kevin Antwi conducted the literature review and contributed to the sections on structural racism and firearm policy. Nia Stewart supported background research and introduction. All authors reviewed, revised and approved the final manuscript.

**Financial support.** This research received no specific grant from any funding agency, commercial or not-for-profit sectors.

**Competing interests.** The authors declare no conflicts of interest.

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
