## [Reviewer Report]

This manuscript addresses a highly relevant and timely issue: the rising rates of firearm suicide among Black and Latino youth in the United States. Its key strength lies in reframing the problem through an intersectional and structural lens, highlighting how systemic racism, inequitable access to mental health care, community trauma, and ineffective firearm policies converge to exacerbate risk. The paper is well written, the argument is coherent, and the call for upstream, equity-focused interventions aligns closely with the scope of Global Mental Health. Still, there are areas where the manuscript could be strengthened to maximize its impact:

Clarity of methodology

- Even though this is an overview review, the manuscript would benefit from a brief description of how literature was selected and synthesized (databases consulted, time frame, inclusion criteria).

- Adding a statement on limitations—such as the U.S.-centric scope, reliance on secondary sources, and lack of systematic search—would improve transparency and strengthen the academic credibility of the piece.

Broader contextualization

- The focus on the United States is justified, but the paper could acknowledge more clearly how these findings relate to international contexts.

- Short comparative references to other discriminated groups globally (e.g., Indigenous populations in Canada and Australia, Roma youth in Europe, refugee youth in different regions) would show that the structural lens has applicability beyond the U.S.

- Highlighting similarities in structural drivers (racism, discrimination, socioeconomic exclusion) despite differences in means of suicide (firearms in the U.S., pesticides in South Asia, hanging in Europe) would enhance relevance for a global audience.

Integration of global frameworks

- The manuscript would be enriched by explicitly situating its findings within existing international strategies, such as WHO’s LIVE LIFE suicide prevention framework and global social determinants of health approaches.

- This would strengthen the connection between the U.S. case study and broader public health debates, increasing the paper’s value to readers outside the U.S.

Commercial determinants of health

- An important dimension missing from the analysis is the role of the firearms industry as a commercial determinant of health.

- The influence of lobbying against gun control, marketing strategies targeting minority populations, and industry-driven normalization of firearms could be discussed alongside structural racism and segregation.

- Drawing parallels with how commercial determinants have been studied in other domains (e.g., tobacco, alcohol, ultra-processed foods) would position the manuscript within a growing body of global literature and expand its interdisciplinary appeal.

Depth of results and synthesis

- While the narrative effectively describes multiple systemic barriers, at times the results feel heavily descriptive. Strengthening synthesis—for example, by organizing findings around a clearer conceptual framework (e.g., socio-ecological model, intersectionality in practice)—could sharpen the contribution.

- Tables or summary figures highlighting the different layers of determinants (structural, institutional, cultural, individual) might help readers navigate the complexity.

Discussion and implications

- The discussion could be expanded to reflect on the transferability of lessons beyond the U.S., including how structural and systemic approaches could inform suicide prevention efforts in other regions, even with different dominant means.

- Attention to community-based and culturally tailored interventions is well presented, but more explicit linkage to international models of care (e.g., community mental health approaches in LMICs, culturally adapted interventions in Indigenous communities) would further strengthen the argument.

- The paper could also reflect more critically on the potential barriers to implementing upstream interventions in contexts where political will, resources, or community trust are limited.

Conclusion

- A conclusion could be enhanced by more explicitly stating how the U.S. case provides lessons for global mental health practice and policy.

- Including a reflection on how commercial and structural determinants interact, and why they must be considered together in future research and interventions, would give the paper a stronger take-home message for an international audience.

---

## [Reviewer Report]

The article is well written and addresses a highly relevant issue: the structural and social determinants of firearm suicide among minority youth.

The scope of the topic is clearly defined, and the paper effectively highlights the importance of the issue being discussed.

The abstract is concise and covers the key aspects of the study. The introduction and literature review provide a solid background, supported by pertinent and up-to-date references.

The authors successfully define and discuss several critical structural factors, including social contagion, residential segregation, community violence and trauma, and barriers to mental health care.

Overall, this is a well-developed narrative review that thoughtfully examines the key structural and systemic contributors to youth suicide. Moreover the address solutions and an signal an action plan including aspects of school policies, law enforcement etc.

---

## [Reviewer Report]

The article by Rhodes et al. addresses a highly important and timely topic. The article is well-written and there are ample up-to-date references. The following suggestions may further strengthen its impact and alignment with the journal’s aims:

First, to better situate the article within global research, the paper should more explicitly contrast the US-specific context of firearm access and cultural factors with global patterns. While systemic contributors such as poverty, discrimination, and structural racism elevate suicide risk among marginalized youth worldwide, the association between firearms and suicide among Black and Latino youths is largely unique to the United States. Emphasizing this distinction would clarify the article’s relevance to global mental health and could help inform discussion in the implication section given that firearm suicide among Black and Latino youths is not observed at similar rates outside North America.

Second, the article would benefit from some discussion of rural versus urban differences in firearm suicide rates among Black and Latino youths, given that these setting differences in firearm suicide have been examined in the literature. Whether the magnitude and drivers of these disparities differ by race and ethnicity, and further exploration of these patterns, could inform targeted prevention strategies.

Third, the article should expand on the possible reasons for the 6% decrease in firearm suicide among White youths, as understanding these factors may inform risk reduction strategies for Black and Latino populations.

Fourth, the article would benefit from minimizing editorializing language throughout the main text. Reserving subjective descriptors (e.g., “grim,” “especially concerning”) for the implications and conclusions sections allows the data and literature to speak more authoritatively, enhancing scientific credibility.

Regarding minor points:

1. The use of “resilience” in the abstract may be inconsistent with the article’s focus on structural determinants.

2. Updating the epidemiological statistics on the proportion of Latino and Black psychologists (line 245) with the most recent data would improve accuracy.

3. References are missing for lines 325-334. It is important that all claims are supported by appropriate citations from the medical literature.

---

## [Editor Report]

Please address reviewers' comments and my own (see below):

1. Clarify the causal pathways linking structural racism to firearm suicide risk

The manuscript effectively documents multiple domains of structural racism (segregation, policing, underinvestment, trauma exposure), but the causal pathways between these mechanisms and firearm suicide risk in Black and Latino adolescents could be articulated more explicitly. A conceptual framework or figure summarizing how these structural factors drive differential exposure to violence, access to firearms, mental health burdens, and ultimately suicide risk would strengthen coherence and help readers understand the multi-level relationships described across sections.

2. Strengthen the distinction between firearm suicide trends and firearm violence trends

While the manuscript provides a robust epidemiologic overview, the early sections sometimes conflate trends in firearm violence (homicide, community violence) with trends in firearm suicide. Clarifying the differences in risk factors, ecological determinants, and temporal patterns for suicide versus homicide—while still emphasizing their shared structural roots—would improve precision. This is especially important because firearm suicide and homicide trends often diverge by race, age, and geography.

3. Address data limitations and variability in estimates

The text cites many individual studies with varying designs, time periods, and geographic scopes. A brief acknowledgment of the limitations of available data—such as underreporting of suicide attempts, misclassification of suicides vs. homicides, and limited granularity in race/ethnicity categories—would strengthen transparency. Readers would also benefit from a short explanation of why some numbers (e.g., nearly 100% increase from 1998–2018) appear extremely large compared to standard CDC/WISQARS reporting.

4. Consider integrating gender differences and intersectionality

Although the manuscript references young men of color and social norms about emotional expression, a more systematic treatment of gender differences in firearm suicide risk would be valuable. For instance, boys/young men have markedly higher rates of firearm suicide compared to girls/young women, and gender interacts with race, immigration status, and acculturative stress in complex ways. This addition would deepen the structural analysis and reflect emerging research emphasizing intersectional risk.

5. Improve organization and reduce redundancy for readability

The manuscript is rich and comprehensive but quite long, with repeated discussions of segregation, trauma exposure, and community violence across different sections. Streamlining overlapping content and reinforcing clear transitions between Level 1 and Level 2 headings would improve flow. Some paragraphs—particularly in the “Community Violence and Trauma” section—cover multiple themes (social contagion, developmental neurobiology, segregation history) that may be more effective if separated or reorganized.

---

## [Editor Report]

Dear Authors,

Thank you for your careful and thorough revision of your manuscript. I appreciate the substantial effort taken to address the comments from both the reviewers and the editorial team.

The revised manuscript demonstrates clear improvements across several key areas. In particular, the articulation of the conceptual framework and causal pathways has been significantly strengthened, the distinction between firearm suicide and other forms of firearm violence is now clearly delineated, and the discussion of data limitations is appropriately nuanced. The integration of gender and intersectional perspectives, as well as the improved organization and flow of the manuscript, further enhance its clarity and contribution.